# Modeling Neurotransmission: Computational Tools to Investigate Neurological Disorders

**DOI:** 10.3390/ijms22094565

**Published:** 2021-04-27

**Authors:** Daniela Gandolfi, Giulia Maria Boiani, Albertino Bigiani, Jonathan Mapelli

**Affiliations:** 1Department of Biomedical, Metabolic and Neural Sciences, University of Modena and Reggio Emilia, Via Campi 287, 41125 Modena, Italy; daniela.gandolfi@unimore.it (D.G.); giuliamaria.boiani@unimore.it (G.M.B.); albertino.bigiani@unimore.it (A.B.); 2Center for Neuroscience and Neurotechnology, University of Modena and Reggio Emilia, Via Campi 287, 41125 Modena, Italy

**Keywords:** synaptic transmission, computational modeling, synaptic plasticity, neurological disorders

## Abstract

The investigation of synaptic functions remains one of the most fascinating challenges in the field of neuroscience and a large number of experimental methods have been tuned to dissect the mechanisms taking part in the neurotransmission process. Furthermore, the understanding of the insights of neurological disorders originating from alterations in neurotransmission often requires the development of (i) animal models of pathologies, (ii) invasive tools and (iii) targeted pharmacological approaches. In the last decades, additional tools to explore neurological diseases have been provided to the scientific community. A wide range of computational models in fact have been developed to explore the alterations of the mechanisms involved in neurotransmission following the emergence of neurological pathologies. Here, we review some of the advancements in the development of computational methods employed to investigate neuronal circuits with a particular focus on the application to the most diffuse neurological disorders.

## 1. Introduction

The mechanisms underlying neurotransmission span over different spatial and temporal scales encompassing multiple fields of research. A multidisciplinary approach is therefore required to investigate one of the most intriguing processes in neurobiology. At the same time, the improved performances of computing platforms boosted the development of mathematical tools to investigate neurobiological mechanisms. The simulation of neuronal functions requires computational models that are based on solid assumptions regarding the mechanisms underlying the analyzed observables. The models are then calibrated to faithfully describe synaptic and neuronal mechanisms to generate reliable predictions. These operational steps are dependent on the available experimental data that in turn allow a fine tuning of model’s parameters.

Synaptic connections are extremely heterogeneous and diversities reflect, among others properties, (i) the molecular specificity of neuronal types, (ii) the coupling between neurotransmitters and receptors and (iii) protein to protein interactions. Furthermore, the huge amounts of neurons and synapses in the brain give rise to structural organizations that need to be further investigated to be reproduced in silico. Beside morphological concerns, the temporal dynamics of the system has also to be taken into account. Synapses in fact can change their functional and structural properties on different time scales, giving rise to short and long-term plasticity. Finally, neurotransmission is governed by stochastic mechanisms like vesicle release or ionic channels dynamics that introduce noise expanding the range of information transfer. Modeling neurotransmission, therefore, requires some reductionist assumptions allowing us to simulate neuronal activity in different behavioral conditions. In this article, we discuss how different computational approaches can be applied to the investigation of synaptic functions and dysfunction.

Computational neuroscience combines two types of methods: pure simulations and machine learning on discovery data. While the latter provides the context for generating simulations through data analysis [1] allowing us to extensively explore databases and bibliography, computational models imply the use of equations describing changes of the system under investigation. These equations are solved by means of numerical simulations to predict the evolution of the system. The main difference between the two approaches is that computer modeling produces large amounts of virtual data complementing experiments [2] rather than analyzing existing data. Although a broad view encompasses a strategy integrating machine learning and simulations, here we focus on purely computational approaches employed to investigate diseases at the level of neural circuits.

### Modeling Neurotransmission

In a broad definition, computational and mathematical models have been proposed to investigate distinct levels of analyses: (i) a computational level (what and why), (ii) an algorithmic level (which) (iii) and a physical level (how), representing respectively brain computation, algorithms and their implementation [3,4,5]. Beside the levels of interpretation, models are often categorized as top-down or bottom-up depending on the scientific issue under investigation. While top down models take phenomena from the computational level and typically remain descriptive, bottom-up models acquire understanding of the rise of a behavior from computations starting from a physical substrate. The integration of these approaches allows to explain the emergence of physiological behaviors, to make predictions on the rising of disorders and eventually to propose therapeutic and pharmacological strategies. In this scenario, biologically realistic models are widely employed to describe at fine grain the behavior of neuronal circuits through differential equations, which are discretized in space and time using finite difference approximations. In most cases, these models are based on Hodgkin and Huxley equations or, alternatively, on the compartmentalization of neuronal morphology. The increased level of abstraction gives rise to “event-driven” models whose most known examples are the spiking networks.

An important aspect of computational modeling is related to the morphological characteristics of neurons and synapses. Among the huge variety of mammalian cells, neurons in fact have different compartments showing functional specificity coupled with molecular and structural peculiarities. Dendrites collect information from other neuronal fibers while soma and axonal processes integrate incoming inputs to elaborate output responses. These operations are dynamically mediated by synaptic activity. It is, therefore, mandatory to generate a reliable representation of the elements participating in neurotransmission to faithfully reproduce neuronal functions.

## 2. Compartmental Modeling

### 2.1. Dendrites

After more than a century from their discovery, dendrites still capture the attention of neuroscientists being the main actors in the play of neuronal communication. The key principles of dendritic operation can be inferred from the integration of predictions and experimental evidence. Computational models allow us to manipulate dendritic properties such as morphology or protein distribution, which are hardly manageable in living tissues. The modeling of dendritic functions lays its foundations in the pioneering cable theory [6] in which the passive properties of neuronal membranes are responsible of the relationship between the depolarization generated by combined inputs and their distance from the soma, which has been indeed observed experimentally [7]. Dendrites most often are quite complex from the morphological point of view and contain several voltage dependent ionic channels enabling neurons to perform complex operations including dendritic spikes and synaptic plasticity (See Figure 1). These characteristics provide dendrites a range of computational tasks such as spatial or temporal coincidence detection [8,9] resembling neuronal networks. The analysis of brain functions during the last two decades revealed a crucial role for dendrites in a wide range of integrative and behaving tasks such as perception [10], sensory-motor integration [11,12] or spatial navigation [13,14,15]. It is, therefore, fundamental to develop innovative experimental strategies to fully dissect the contribution of dendrites in neuronal computation [16].

Dendritic models can be significantly different in the level of realism and in the complexity. It is therefore difficult to identify the best model to describe dendritic functions, as it depends on the question to be answered. Computational models often use morphological changes to assess neuronal responses [17,18,19,20]. For instance, models were used to investigate the correlation of firing activity with the size of the dendritic branching in different brain areas [21,22]. Large dendritic trees were predicted to generate low bursting rates, high threshold and a lower tendency to induce bursts [21]. Similarly, the dendritic length, the volume and the number of branches contribute to discriminate from regular to bursting firing [22]. In detailed models of pyramidal neuron, the effects of dendritic bifurcation on signal integration were predicted to affect the ability of synaptic input to generate and propagate action potentials [23] and neuronal gain modulation [24]. These models allowed us to underlie the relationship between structure and function [25] in dendritic computation.

Beside morphology, the functions of dendritic ionic channels are yet to be fully explored [26]. The modeling of biophysical properties is optimal to investigate how ion channels features influence neuronal activity. In these models, all the functional properties of membrane proteins can be reversibly modulated to verify hypotheses on dendritic functions. It has in fact been possible to reveal the influence of dendritic conductance on synaptic integration, membrane potential changes, signal propagation and synaptic plasticity [27,28,29,30,31]. For example, detailed biophysical models were used to unravel the contribution of potassium channels in the generation of AP backpropagation [32] and in the reduction of AP amplitude [33]. Similarly, models predicted that the abundance of H-channels in CA1 pyramidal neurons can block the temporal summation of not-synchronized distal inputs [34,35].

Moreover, it has been shown that the activity of voltage-gated channel in the postsynaptic neuron can amplify small synaptic input at distal dendrites leading to slow depolarization in the network [36]. Computational models demonstrated to be useful in delineating how conductance, morphology and timing interact to the emergence of coincidence detection [37,38]. According to these models, dendritic spiking in basal dendrites of CA1 neurons results from coincident synaptic inputs leading to a temporal precise generation of somatic AP [39]. Dendritic spikes could therefore serve as timers of somatic APs. A similar behavior has been observed in the apical dendrites of CA1 were the spike initiation requires the synchronous activation of about 50 synapses [37]. Although experiments showed the predicted synchronicity, only computational models were able to infer that patterns of stimulation are strongly involved in the regulation of dendritic spikes initiation. More specifically, the differential role of NMDA and AMPA receptors can extend the temporal window for membrane potential integration [40]. These examples demonstrate how computational models can make predictions such as the one reported for coincidence detection, raising the important issue regarding the possibility to test specific patterns of stimulation, which in most cases cannot be explored experimentally [16,41].

### 2.2. Synapses

Much of the efforts in the analysis of synaptic functions during the last decades has been devoted to investigate long-term synaptic plasticity [42,43], either in the form of potentiation (LTP) or depression (LTD). Synaptic plasticity is crucial in determining several brain processes such as formation and storage of memories [44], attention or receptive fields selection [45]. Furthermore, a wide series of neurological diseases have been correlated with deficits in synaptic plasticity [46,47]. From the computational perspective, reliable models of synapses are required [48] and while synapse is classically viewed as a single compartment acting as a linear multiplier of incoming inputs, learning rules of synaptic plasticity enable the optimization of gain control by Hebbian activity [48,49]. The vast majority of computational models of circuit functions [50,51,52] is based on this simplistic view of synaptic architecture and is efficiently employed for state-of-the-art machine learning algorithms [53] (Figure 2A). Nevertheless, this scheme completely neglects presynaptic functions. In fact, the idea of synapses acting as simple gain controllers fits well with the postsynaptic element whereas the non-linear and stochastic nature of neurotransmitter release is not considered. Extended models, where signals are filtered before the gain control performed by the postsynaptic element, have been recently proposed [54]. Interestingly, postsynaptic site usually shows linear functions whereas the stochasticity of presynaptic release complies with non-linear responses that expand the dynamical range of neural functions. Notably, both compartments are subject to a set of independent learning rules, optimizing their functions to increase the synaptic information processing (Figure 2B).

By adopting this approach, presynaptic compartments behave as frequency filters, allowing terminals to perform temporal filtering that discriminates incoming firing patterns [55,56]. A further source of non-linearity is introduced by the asynchronous neurotransmitter release compared to presynaptic action potential arrivals [57] and that can persist after the last spike [58]. It has been shown in fact that, although neurotransmitter release is predominantly synchronous, some synapses conform to this asynchronous mechanism [59,60]. 

## 3. Modeling Diseases

Given the level of details in describing neurotransmission process, in silico investigation is becoming a tool for preclinical and clinical research to understand brain diseases. In recent years, we have witnessed the development of models providing insights into mechanisms underlying brain diseases [61] and drug action [62]. Computational models require set of equations that describes changes of neuronal functions parametrized on experimental data. This framework, on one side allows to develop and design experimental approaches, on the other side permits to change parameter setting exploring conditions that in most cases can be hardly tested experimentally.

Given the importance of data in the generation of neuronal models [63,64], the simulation of brain disorders has been preferentially tailored onto pathologies that have been extensively explored providing a large amount of experimental data. We have therefore focused our attention on four of the most studied neurological diseases: epilepsy, Alzheimer’s and Parkinson’s diseases and schizophrenia. We have attempted to give an overview of the computational models that have been successfully used to provide insights on the mechanisms related to the emergence of the diseases. 

### 3.1. Epilepsy

Epilepsy is a set of disorders that can involve different brain areas, from cortex to deep regions. Notably, this functional polymorphism is coupled with genetical and pathophysiological heterogeneity that can be encountered at different levels: from ion channels and membrane proteins to the scale of brain wiring [65]. Differently from other brain disorders, such as migraine or ischemic attacks deriving from hemodynamics failures, epilepsy is a dynamical disorder originating from neuronal dysfunction. The main manifestation among epileptic symptoms is seizure, emerging when large brain areas produce highly synchronous and uncontrolled neural activity. This neurological symptom defines epilepsy when occurring spontaneously or recurrently, while it can be easily induced in neuronal networks through pharmacological treatments. For this reason, the experimental investigation of epileptogenic tissues is still the most prominent method to analyze the biophysical mechanisms underlying epilepsy, together with neuroimaging and electrophysiological recordings coupled with genetic mapping of patients. Nonetheless, computational modeling is increasingly becoming a widespread tool to link these fields of investigation. The simulation of a neurological disease such as epilepsy could require to shaping the model directly on the symptoms. Nevertheless, these models would attain a complexity level that could severely limit the simplification required in the definition of computational models. It is, therefore, preferable to adopt a multiscale approach combining the cellular and microcircuit level to scale up to clinical level [66,67]. At the spatial scale, epileptic models range from protein and membrane level to brain regions [68], whereas at the temporal scale the range is very broad: from millisecond to years. Designing a model that accounts for these spatiotemporal scales can be very hard [69], however, there is a rich repertoire of works showing how static [70,71] and dynamic models can be used to model the emergence of seizure [2,72,73]. Notably, the use of deterministic models of archicortical circuits successfully allowed to reproduce a wide variety of patterns that have been observed in patients [74,75].

#### 3.1.1. Realistic Models

The detailed neurobiological reconstruction of neuronal circuits activity is still the most productive area of neural modeling applied to epileptic research.

One of the mechanisms involved in the emergence of epilepsy is the alteration of ion channels dynamics that underlies dysfunctions in neuronal firing patterns [76]. To this aim, biologically realistic models faithfully reproduce neuronal behavior and they can thus be focused on the changes involved in dysfunctions that underly neurological diseases, by alternatively zooming on protein kinetics, molecular pathways or ion channel dynamics. This approach allowed us to explore the emergence of diseases from the disruptions of complex molecular interactions. In particular, the alterations induced by ion channel mutations have been modeled to simulate changes in neuronal excitability induced by epilepsy [77,78]. The application of Hodgkin and Huxley (HH) equations to ion channels kinetics indeed proved a valuable tool to study the effects of epileptogenic mutations. Neuronal models have been used to simulate (i) the up- and downregulation of channels densities and kinetics that have been shown to occur in response to epilepsy [79,80,81]; (ii) the increase of Ih-current observed following seizures that hyper-excite neurons [82]; (iii) the changes of ion concentration gradients during seizures [83]. This latter condition should be indeed emphasized, since HH models revealed the critical role of ion concentration gradient in neuronal activity that leads to changing excitability [84] and network synchronization [85] reproducing a condition that can be hardly reproduced experimentally. Furthermore, while active channels are mainly located on neuronal elements, glial cells, which are not directly involved in the modulation of electrical activity, participate in the regulation and maintenance of ion gradients. For example, simulations performed including glial elements in the circuit demonstrated that changes in potassium diffusion affect the oscillations of intracellular ion concentration that have been detected during seizures [83,86]. This aspect is also important given the strong difficulties of simultaneously measuring the activity of mixed neuronal and glial populations. On the pharmacological side, antiepileptic drugs are often targeted to voltage gated ion channels [87], affecting their functioning [88,89]. Computational modeling proved efficient in the understanding how molecular interactions between drugs and ion channels are involved in determining cells behavior. For instance, a recent modeling framework has been proficiently applied to predict the tyrosine kinase receptor Csf1R as a potential therapeutic target for epilepsy [90]. Alternatively, mathematical simulations were used to disambiguate hypotheses on the action mechanisms of pharmacological molecules either by testing different configurations [87], or by using models accounting for the kinetics of single ion channel [91].

At a larger scale, realistic models of spiking networks, where neurons are reduced to single computational points, showed that population activity can be altered by switching neuronal firing from spiking to bursting [92]. Notably, experimental recordings showed that epileptic activity in hippocampal regions can be triggered by single cell bursting [93]. Similarly, the characteristic seizures, which were shown to depend either on the block of GABAergic inhibition or on the enhanced excitatory connections, could be faithfully reproduced by an increased synaptic excitation [94]. Interestingly, models allowed to infer the correct topology of the hippocampal system since results fitted experimental data only when particular geometrical constraints were predetermined [95]. Network simulations revealed a prominent role for axo-axonic electrical synapses [96] and, more generally, for the network topology and synaptic propagation [97].

An interesting example of the power of computational models can be found in the analysis of the action mechanisms of antiepileptic drugs. The synchrony and rhythmic firing showed by thalamocortical circuits in healthy conditions seems to depend on inhibition [98,99]. However, it is well recognized that one of the most employed categories of antiepileptic drugs is the GABA agonist. The mechanisms underlying this paradox have been investigated by using network models. Simulations revealed that thalamocortical networks lose their synchronization in response to the increase of GABAergic inhibition in the reticular nuclei [100].

A further increase of the network size requires us to reduce the level of complexity of single neuron while preserving dynamics at multiple levels. The easiest way is to combine currents showing similar kinetics into a single unique parameter as in the Morris-Lecar model [101]. This model has been adapted to investigate transitions from tonic to clonic phase during epileptic discharging [102], and to demonstrate how inhibitory interneurons coupled through gap junctions drive the slow transition of global excitability into paroxysmal regime [103]. Additionally, a variant of the classical HH model has been developed to investigate the causes underlying the end of a seizure [104]. By simulating the dynamics of ion concentration, the authors found how sodium, potassium and chloride concentrations regulate the alternance between resting and spiking/bursting activity of a circuit. Interestingly, human epileptic tissues were shown to increase asynchronous GABA release from inhibitory neurons [105], suggesting its importance in balancing pathological hyperexcitability. Although experimental observations allowed to identify the existence of such mechanisms, computational models and simulation proved fundamental to investigate the functional impact of asynchronous neurotransmitter release [106,107,108,109].

Large scale models are also used to investigate the mechanisms leading to signals recorded by EEG and therefore to identify brain states and the relative changes in pathological conditions. In particular, during seizures brain states may undergo changes, which can be detected through EEG recordings.

Large scale models have been proficiently used to study the spatiotemporal pattern emerging during seizures [110], allowing to investigate the spatial extent of seizures between different recording sites of a typical EEG measure or, alternatively, to provide explanation of the pattern of activity generated by stimulating electrodes [111].

The development and application of neural mass models provided an intermediate approach linking the mesoscale to the microscopic scale that have been successfully applied to determine a taxonomy of epilepsy [112] and to explore the mechanisms underlying the alternance between ictal and interictal discharges [113].

#### 3.1.2. Abstract Models

Differently from realistic models, where actual physiology is retained, abstract models largely lack physiological details. Nonetheless, the simplicity of abstract models allows us to extract general relationships between input and output features. Very simple and abstract networks have been used to simulate collective oscillations in epilepsy [114,115] and their changes over time [102]. Since most of the therapeutic approaches attempt to change synchrony, which is believed to derive from epileptic activity, the model can be employed to determine the level of such changes to explore different strategies of intervention.

A major limitation in abstract modeling derives from the difficulty to account for connections in the circuit architecture. Until advanced imaging methods will provide well-resolved data [116,117], the approach based on random connectivity is nowadays employed in abstract models to investigate how small changes in the connectivity matrix can lead to behavioral phenotypes. For example, random networks show epileptiform activity like in cultured neuronal networks [118,119] and network topology can be altered to reproduce non-physiological behaviors by acting on the number of connections [119]. Interestingly, the analysis of network topology suggested the existence of “hub” neurons, receiving a large number of connections that influence the synchronization of large brain regions [120,121,122]. According to this evidence, the connectivity of a neural network critically affects the functional properties of the circuit and may lead to pathological disfunctions [123]. For instance, it has been shown that long-range connections lead to faster information processing and synchronization, whereas a limited number of such connections has been implicated in the emergence of epilepsy [124,125]. In addition, abstract network models proved valuable in highlighting the increased connectivity between electrodes close to epileptic foci [126]. These statistical models can synthetize the complexity of a multivariate system into few salient features.

### 3.2. Alzheimer’s Disease

Alzheimer’s disease (AD) is a neurodegenerative disease compromising cognitive functions and severely affecting the quality of life of patients. AD is a complex and multifactorial disease characterized by the degeneration of neurons and by the disturbances in neuronal synapses within cortical and subcortical areas [127]. From the pathological point of view, AD is characterized by an abnormal accumulation of the beta-amyloid (Aβ) peptide and of the hyperphosphorylated tau protein. Despite the huge efforts over the last decades, the universal cure for AD is still far from being released mainly because a full understanding of the mechanisms governing the expression of AD in humans is still far from being obtained. Among these mechanisms, the accumulation of amyloid plaques and neurofibrillary tangle (NFT) have been reported to be involved. Several hypotheses have been put forward to explain the causes and symptoms of the disease: (i) The cholinergic hypothesis considers a reduction in the acetylcholine (ACh) cycle leading to an overall loss of cholinergic innervation in the cerebral cortex [128]; (ii) the amyloid (Aβ) hypothesis envisages a role for the Aβ peptide in the formation of plaques [129]; (iii) the Tau hypothesis states that the abnormal phosphorylation of tau protein leads to the formation of NFTs that are massively present in AD brains [130]; (iv) the glucose synthase kinase 3 (GSK3) hypothesis states that the over-activity of this kinase accounts for most of the mechanisms leading to AD symptoms such as memory impairment [131]; (v) other hypotheses like cerebrovascular disease and inflammation have been proposed and extensively explored [132].

Multiscale and multilevel modelling approaches attempted to provide new insights into AD mechanisms and can be further expanded to identify novel therapeutic targets and pharmacological treatments such as the serotoninergic system, which has been deeply investigated through computational methods over the last decade [133].

#### 3.2.1. Realistic Models

Among the hypotheses that have been extensively investigated, the Aβ hypothesis is the most widely applied to computational models of Alzheimer’s disease. The Amyloid-beta (Aβ) is a peptide that is believed to be causative in the disease process since in brains of patients with Alzheimer’s disease its level is elevated and it forms characteristic aggregates, the plaques [134]. The Aβ increase reduces glutamatergic transmission and inhibits synaptic plasticity. There is an intense debate on the genesis of the Aβ pathways that has yielded to the development of multiple approaches to investigate this issue, both with experimental [135] and theoretical methods [136]. Computational models have been generated to understand the biochemical basis of self-association of Aβ peptides [136,137]. Among others, in 2001 Pallitto and Murphy [138] made significant improvements by developing a model featuring experimental data on the Aβ formation including (i) the growth mechanisms, (ii) the monomere addition of (iii) fibrils and (iv) filaments, (v) the mass fraction and fibril length. A few years later Kim and colleagues [139] examined how Aβ oligomers are controlled by urea. Other modeling approaches are focused on the mechanisms of plaques formation [140], on the kinetics of their formation [141] and on the interaction between intracellular calcium and Aβ [142]. Other biochemical pathways have been explored through computational methods in order to predict some of the clinically relevant outcomes of AD. Anastasio in 2013 [143] proposed a model in which the Aβ formation is related to cerebrovascular disease, inflammation and oxidative stress. More recently, models have been developed attempting to simulate biochemical signaling related to TNFα, whose dis-regulation triggers hyperphosphorylation of tau protein leading to a massive glutamate release inducing microglia activation and neuronal death [144]. Alternatively, the reciprocal interaction between different biomarkers has been modeled to predict the emergence of Aβ plaques and the subsequent symptoms [145].

At the cellular and microcircuit level, detailed biophysical models of Aβ plaques formation allowed to investigate the involvement of hippocampal circuits in the expression of AD. Notably, these models have evidenced a prominent role for transient A-type K^+^ currents in the changes on firing patterns caused by Aβ accumulation at somatic, synaptic [146] and membrane level [147].

Furthermore, realistic models have been used to analyze how the prolonged exposure to Aβ affects ionic conductance [148]. For example, simulations on a hippocampal CA1-medial septum network showed that A-type K^+^ channels inhibition induced the theta-band rhythm increase observed in EEG recordings. Similarly, cortical oscillations observed in patients with Alzheimer’s disease were reproduced through network models highlighting that theta band power is affected more than other oscillatory bands by excitatory activity and synaptic loss following Alzheimer’s disease [149].

Given the importance of Aβ peptide in altering glutamatergic transmission, realistic models of cortical circuit simulating working memory were used to analyze the effect of NMDA on the disease [150]. The pathological condition was modeled by a reduction of cholinergic tone coupled with diffuse synaptic and neuronal losses. Similarly, advanced models of hippocampal network dynamics [151,152] attempted to explain the memory loss and cognitive decline of Alzheimer’s disease through firing activity changes in hippocampal CA3 neurons mediated by a reduction of cholinergic input.

Realistic models of CA1 have been also used to explore the effects of CREB (modeled as a decrease in AHP conductance) on neuronal and synaptic activity. Interestingly, these models were used to infer that the use of CREB-based therapies could provide a new approach to AD treatments [153].

#### 3.2.2. Abstract Models

Abstract models of large neuronal networks have been adopted to explain how the direct brain stimulation could contribute to slow the progression of the disease [154]. Moreover, several variants of abstract networks have been proposed to investigate the mechanisms underlying the memory loss typical of AD [155,156], with a particular attention on hippocampus and parahippocampal cortices, either by simulating the impairment of the functionality of a particular area [157] or the relationship between neural elements with the endocrine system (e.g., cortisol [158]). It should also be noted that, since the early 1990′s, system level models are used to simulate Alzheimer’s disease [159] and proved valuable to mimic the memory deficit in patients. The implementation of Hebbian learning rules in feedforward neural network and the selective alterations of groups of neurons [160] have been proficiently employed to simulate the cognitive deficits and the memory impairments that are typically encountered in all patients. More recently, Hopfield networks, an artificial neural network characterized by recurrent connectivity, have been used to demonstrate that the random deletion of synaptic weights leads to memory impairments [161]. Noteworthy, the salient details on EEG activity can be estimated by using graph theory applied to the analysis of large-scale abstract models in which electrodes are represented by nodes and the strength of the coupling among nodes is the edge between links [162,163]. An interesting attempt to understand key protein/drug interactions at the system comes from the development of abstract probabilistic (Bayesian) network models [164] which allowed to assess the biological connectivity between different AD associated blood-based proteins.

### 3.3. Parkinson

The pathology underlying Parkinson’s disease (PD) is a degeneration of dopaminergic neurons in the substantia nigra, leading to the decrease in the level of dopamine (DA) in the dorsal striatum especially the putamen. Among the symptoms characterizing PD, tremor, slowness of movements and more generally motor impairments are the most widespread. Unfortunately, also non-motor symptoms can appear during the progression of the disease. The majority of computational models employed in the field of PD have been generated to investigate the mechanisms underlying the disease and to provide theoretical support to treatments such as Deep Brain Stimulation [165]. Although the loss of dopaminergic neurons has been linked to PD and the main circuits have been determined, several studies have shown that symptoms associated to PD such as akinesia or resting tremor may involve different brain circuits and neural processes. Beside the typical striato-thalamo-cortical circuit, the mesolimbic pathway and the cerebello-thalamo-cortical circuit have been suggested to be involved in the expression of the disease [166]. Parkinsonian symptoms are associated with the degeneration of the dopaminergic system, however, the exact nature of the impairments is not yet fully understood. The use of advanced computational methods encompassing different levels of analyses was shown to contribute to the understanding of such mechanisms.

#### 3.3.1. Realistic Models

Several realistic models have been developed to investigate the mechanisms leading to the loss of dopaminergic neurons. At the molecular level, one of the most recent theories suggests that an energy deficiency at sub-cellular level can cause the cell loss in the substantia nigra in PD. The development of a comprehensive computational model of the neurons in the substantia nigra suggested a more crucial role for hypoglycemia in ATP deficit rather than hypoxia [167]. The loss of midbrain dopaminergic neurons that alter dopamine dynamics have been explained by tailoring kinetics models of dopamine release mimicking the interplay between dopamine release and re-uptake [168,169]. The loss of dopamine terminals observed in PD patients causes a reduction in neurotransmitter release and in the uptake. Another important aspect of modeling synaptic transmission is related to neurotransmitter diffusion [170], which only recently has been modeled with various levels of complexity [171,172]. Wiencke and colleagues proposed a computational model for dopamine transmission integrating diffusion, uptake and release from multiple terminals. The authors argued that the simulation of dopamine concentration in synaptic, peri-synaptic and extra-synaptic compartments can be employed to mimic pharmacological manipulations or dopamine-related disorders [171]. In PD, motor deficits are known to occur as a major consequence of dopamine depletion (DD) in the striatum. Modeling single projections neurons has been necessary to understand the interplay between DD, dopamine receptors up- and downregulation and changes in neuronal excitability [173,174,175]. These models showed that PD changes the balance of direct and indirect pathways of activation on a biophysical basis [176,177].

The well-defined circuits involved in the expression of PD allowed the creation of computational models to simulate the network involved in PD such as the cortex-thalamus-basal ganglia circuit, the striatum circuit or the globus pallidus circuit. Some of the electrophysiological features changing in response to PD have been well characterized, for instance increased firing rates, bursting regime, interneural synchrony and oscillatory activity [178]. Given these well-known outcomes of the disease, models are generally focused onto specific mechanisms to investigate their contribute to the expression of PD. As in other cases, PD has been investigated by means of network models. By using a whole striatum circuit, it has been shown that DD increases the level of correlation between neurons firing [179]. Alternatively, DD was shown to change the output of inhibitory interneurons altering the balance of projections neurons activity [180]. At a larger scale, simulations over the whole basal-ganglia network were used to analyze the effects of DD in the striatum. These models showed that DD disrupts the balance of the two striatal output [181,182]. Conversely, DD was suggested to disrupt both striatal pathways [183]. In all cases, these models predicted motor deficits related to a disruption of the basal ganglia functions due to the loss of inhibition. An interesting analysis on the power of computational tools to investigate neurodegenerative diseases comes from the work of Humpries and colleagues [184] reporting the case study of the PD with a focus on the causative mechanisms, the neural dynamics and the treatments. This aspect is particularly important for facing PD since motor symptoms are typically accompanied by a massive power in the alpha-beta regime (7–35 Hz) together with a large synchronization of neurons encompassing the cortex-basal-ganglia-thalamus network. The increased synchrony observed in the EEG activity can be successfully treated with deep brain stimulation (DBS). Recent models have been proposed to investigate new targets for less invasive therapies by elucidating mechanisms supporting DBS [185,186]. Importantly, with the aim of developing novel DBS methods and optimizing the effects of DBS on patients, computational methods demonstrated that an increase in the DBS stimulation frequency tends to produce action potentials with higher variability and reduces significantly the alpha-beta power in almost all brain nuclei.

#### 3.3.2. Abstract Models

As in the case of other diseases, abstract models can be very useful in analyzing the mechanisms underlying the emergence of PD symptoms. Furthermore, such models have been proficiently employed to develop the treatment protocols of DBS defining the stimulation patterns to be delivered to electrodes in order to dampen the motor consequences of PD [187,188,189]. In particular, it has been proposed that synchrony, rather than firing rate, is the critical pathological feature of PD symptoms [188]. Abstract models have been used to propose periodic stimulations as therapeutic protocol. Periodic stimulations could in fact induce chaotic desynchronization by interacting with the intrinsic oscillatory mechanism of globus pallidus neurons, remarkably limiting the symptoms of PD. In a similar framework, abstract models have been adapted to simulate the dynamics of the cerebellar-basal ganglia-thalamocortical network, which oscillate in the frequency domain at a value relevant for movement disorders [190]. These oscillations are an effect of network dynamics where each region shows different kinetics and can transit from an oscillatory regime to another in dependence on DBS.

The synchronous oscillations appearing as a key pathological feature of PD are still object of debate regarding their origin. Mean-field models, which describe the collective dynamics of large networks base on interacting spiking neurons, have been applied to explore the onset mechanisms of PD.

### 3.4. Schizophrenia

Schizophrenia is a major mental illness impacting patients and their social interactions, presenting a huge level of heterogeneity in symptoms from cognitive disabilities to isolation and hallucination [191]. Schizophrenia manifests itself through various symptoms across patients, which can be classified as positive, negative and cognitive symptoms where positive symptoms can either be hallucination or delusions, negative symptoms include apathy and reduced affectivity while cognitive deficits relate to impaired attention and reasoning and a decreased memory performance, usually associated with a loss of several points from the IQ score [3]. Despite a poor understanding of the causes and mechanisms of the disorder, several studies have identified that an interaction between genetic and environmental risk factors are necessary to the emergence of schizophrenia [3].

In past decades, researchers have identified some differences between patients and healthy subjects that allowed them to make some hypotheses on the origins of the disorder. (i) The dopamine hypothesis derives from the findings that the dopamine receptor D2 is the target for the most widely employed anti-psychotic drugs. This hypothesis has been then confirmed with neuroimaging and molecular analysis revealing elevated levels of dopamine receptors in striatal and pre-frontal regions of patients [192]. (ii) The glutamate hypothesis derives from the observation that healthy subjects suffer from psychosis following the exposure to psychoactive drugs blocking NMDA receptors activity [193], suggesting that NMDA hypo-functioning can account for some of the observed symptoms. (iii) The GABA hypothesis derives from the reduced GABA observed in pre-frontal regions of patients [194]. (iv) The disconnection hypothesis derives from the observation of abnormal morphological properties (connections, folding, ventricle volume, synaptic connectivity) which have been supposed to result from a reduced synchrony between cortical areas [195].

Computational models have started to address the causes of divergence in clinical symptoms. In particular, despite the complexity of the system, biologically plausible networks are now becoming an efficient tool for the investigation of this illness [3].

#### 3.4.1. Realistic Models

Most of the models exploring mechanisms underlying schizophrenia rely on integrate and fire neuronal networks. The use of circuits made of Hodgkin and Huxley or simpler IF neurons allowed researchers to show how changes induced by dopamine on ionic and synaptic currents influence the stability of strongly interconnected networks [196,197]. These works lead to hypothesize the existence of two alternative states dominated by the activity of dopamine D2 and D1 receptors [198]. These states might be related to the different symptoms observed in schizophrenic patients. Moreover, given the importance of NMDA currents in the induction of plastic mechanisms, computational models have hypothesized that the up- or downregulation of NMDA activity could contribute to induce the two alternative network states correlated with schizophrenic symptoms [199,200]. The level of dopamine could therefore cause cognitive deficits by modulating NMDA currents [201]. The dopamine hypothesis has been further explored by implementing realistic networks endowed with AMPA, GABA, NMDA and dopamine pathways. In one of the proposed models, the authors support the statement that cognitive decline descends from a reduction of the signal-to-noise ratio in cortical neurons which is modulated by dopamine activity through a stabilization of the firing patterns [201,202]. Similarly, the glutamate hypothesis has been investigated by simulating cognitive tasks typically involved in the symptoms of schizophrenia through networks modeling the interactions between excitatory and inhibitory activity in cortical areas. In some works, by simulating NMDA hypo-function either in prefrontal cortex [203] or hippocampus [204], realistic models proved capable of predicting memory impairment compatible with clinical symptoms. The realistic models exploring the GABA hypothesis are often integrated into simulations encompassing NMDA hypofunction. These models suggest that a deficient inhibition could lead to the disruption of rhythmic cortical activity as observed in schizophrenia [205]. Interestingly, the predictive capability of these models is very detailed despite the difficulties in obtaining experimental data. Most of the measurements are acquired in fact through neuroimaging or behavioral experiments and therefore hardly relate to precise neuronal mechanisms.

NMDA receptor hypofunction was also described in a recent (neural mass model derived as an exact mean-field) model of a network of synaptically coupled spiking neurons, accounting for abnormality seen in primary motor cortex, where a sharp decrease in neural oscillatory power in the beta band is observed during movement followed by an increase above baseline on movement cessation [206].

#### 3.4.2. Abstract Models

The analysis of clinical symptoms in schizophrenia is critical and a comprehensive view of all the aspects of the disorder can be hardly explored by means of realistic spiking networks. Typically, fMRI and EEG experiments are used to deepen the impact of circuit alterations on behavioral functions. Abstract models have been also proposed to explore the symptoms and the mechanisms underlying schizophrenia [207,208]. By using this approach, it was proposed that synchrony between large cohort of neurons could entrain stable state in the network avoiding the intervention of distracting inputs. Moreover, computational models of large networks attempted to explain cognitive symptoms of schizophrenia through a decline of the signal-to-noise ratio (SNR) of cortical neurons [3]. An interesting approach in the analysis of schizophrenia is also provided by neural mass model where the dynamics of a brain region, for instance a node in brain network is described by neural mass [209]. Schizophrenia has been in fact associated with a reduced suppression of activity in the default-mode network if subjects are engaged in external tasks. Under conditions of near-criticality, the spontaneous formation and subsequent propagation of neural cascade throughout the network characterize schizophrenia.

Finally, there is increasing interest in Bayesian models of neural circuits [210,211]. Several research groups are working with these models, which have been proficiently employed to investigate various neural disorders. Notably the dynamic causal model (DCM, [212]), a statistical framework initially developed for fMRI analysis has been applied to investigate large scale brain regions in physiological and pathological conditions [213]. The two most recent and intriguing hypotheses on schizophrenia have been indeed investigated by means of large-scale computational models: the disconnection and the Bayesian inference hypothesis. On one side, simple Hopfield networks [214] and three layers perceptrons [215] were used to mimic pruning of synaptic connections. On the other side, according to his proponent [195], brain evolved to interpret and infer the causes and consequences from environment to predict outcomes by minimizing surprises and schizophrenia could be linked to a Bayesian inference deficit [216].

## 4. Conclusions

The parallel progresses in the experimental and computational Neuroscience, together with the improvement of high-performing computing prelude to an exponential increase in the development of computational models aiming to investigate the mechanisms underlying neurological diseases (see Figure 3). In the near future, we can foresee that researches and scientists’ efforts will be increasingly devoted to the generation of computational models in order to provide effective therapeutic and pharmacological strategies.

## Figures and Tables

**Figure 1 ijms-22-04565-f001:**
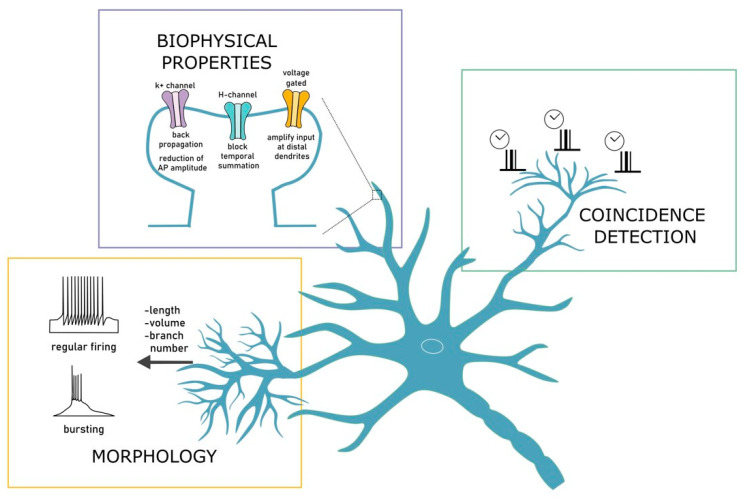
Dendritic modeling. Dendrites are the main participants in neuronal communication. Their morphological, biophysical and integration properties influence neuronal activity. Morphology, in terms of dendritic length, volume and branch number, contributes to discriminate from regular to bursting firing, and influences functional properties as bursting rates, threshold and tendency to induce bursts. Biophysical properties reveal how ion channels features influence, for example, synaptic integration, membrane potential changes, signal propagation and synaptic plasticity. Biophysical models helped to unravel the contribution of channels in the modulation of AP generation and temporal properties. Finally, conductance, morphology and timing interact to the emergence of coincidence detection, making dendrites serve as timers of somatic APs.

**Figure 2 ijms-22-04565-f002:**
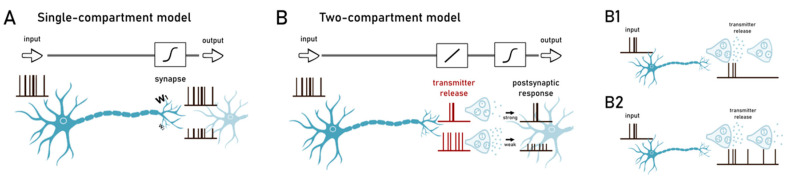
Synaptic modeling: (**A**) Single compartment synapse acting as a linear multiplier of incoming input. Despite its simplicity, this model lies at the basis of a wide range of computational models of circuit function. (**B**) Two-compartment models. Presynaptic terminal is believed to behave as a frequency filter able to perform complex forms of temporal filtering discriminating firing patterns while the postsynaptic element controls the gain to modulate neuronal response. (**B1**) Another level of complexity in modeling synaptic transmission is related to neurotransmitter diffusion. Here synchronous release of neurotransmitters is represented; (**B2**) asynchronous release of neurotransmitters.

**Figure 3 ijms-22-04565-f003:**
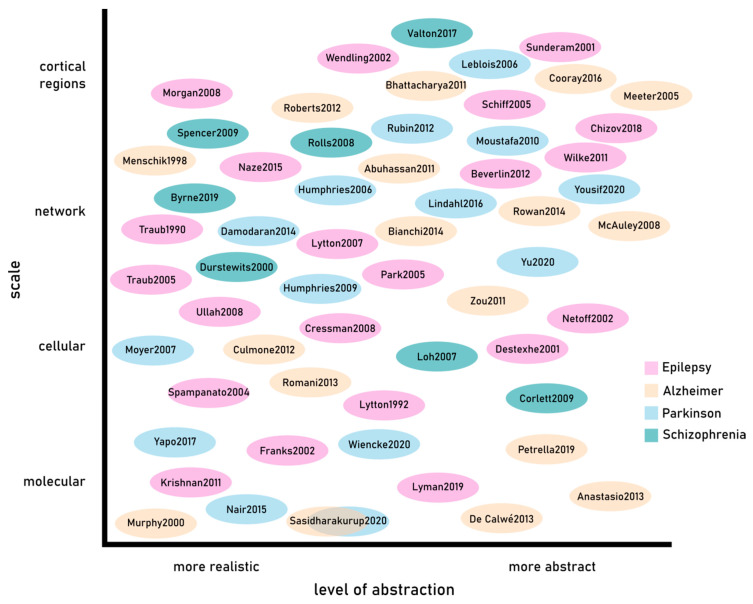
Computational models.

## Data Availability

Not applicable.

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
