# Peer review of "Modeling Neurotransmission: Computational Tools to Investigate Neurological Disorders"

_ijms, 2021, doi:10.3390/ijms22094565_

Round 1

Reviewer 1 Report

In this manuscript the authors review the recent advancements in the development and in the application of computational approaches to explore the functioning of neural circuits with a particular focus on neurological disorders.

Writing a review usually require a big effort in literature search and analysis and it cannot include all published literature. However, in this manuscript, few space has been given to Parkinson's disease, while a bigger effort has been done, for example, in collecting literature on epilepsy. Therefore there is an imbalance in the treatment of topics that must be compensated. Moreover treating all neurological disorders at the same time, by dividing the Sections depending on the spatial scale they treat, generate a confusing mixture of topics. Finally, even thought it could not be considered as a focus of the review, I think it is necessary to better explain the physiological problems and causes related to the different neurological disorders and to better explain the hypothesis underlying the different cases. Only introducing the different neurological disorders in a wider framework, it will be possible to follow the presented state-of-art.

In the following I list my main questions and remarks. To summarize my report, I think that a substantial revision is necessary in order for the manuscript to be published.

Major remarks
-Sec. 3.1 "Molecular level models".
The first paragraph on the Aβ hypothesis for modelling Alzheimer disease is sloppy and unclear. Since the manuscript is meant to be a review, this section should give a brief introduction on the problem and on the hypothesis. At the moment it looks like a list of papers on a certain (not introduced) hypothesis, that is readable just for experts.
The same can be said for the next paragraph on the loss of dopaminergic neurons in Parkinson's disease, while a more consistent introduction has been written for epilepsy.

-Sec. 3.1 "Molecular level models".
I would suggest to divide the section in sub-sections, each one taking into account a single disease case. Equilibrate the 3 cases (Parkinson's disease is barely treated at the moment) and separate them for a better understanding.

-Sec 3.2 "Realistic circuit models"
Please clarify the differences between the models proposed here and in the previous section. Which scales are considered in the different models? which dynamics do the different models want to reproduce?

-Sec 3.2 "Realistic circuit models"
As before, I would suggest to divide the section in sub-sections, each one taking into account a single disease case. Equilibrate the 3 cases (epilepsy disease is barely treated at the moment) and separate them for a better understanding.

-Sec. 3.3 "Neuronal networks"
In this section theta-neuron models have been briefly cited as possible example of neuron models. However a flourishing literature has emerged in recent years starting from this model, that has been completely ignored in this manuscript. Starting from the theta-neuron model, a next generation neural mass model has been recently developed in:
Montbrio, E., Pazo, D., and Roxin, A. (2015). Macroscopic description for networks of spiking neurons. Phys. Rev. X 5, 021028

The exact reduction dimension techniques at the basis of the next generation neural mass model have been developed for coupled phase oscillators and allow for an exact (analytical) moving upwards through the scales. Therefore, with these next generation neural mass models it is possible to develop an intermediate "across-scale" approach exploiting the 1:1 correspondence between microscopic and mesoscopic level. 
These models have been recently applied (among many cases) to modelling electrical synapses [Montbrio, E. and Pazo, D. (2020). Exact mean-field theory explains the dual role of electrical synapses in collective synchronization. Phys. Rev. Lett. 125, 248101], working memory [Taher, H., Torcini, A., and Olmi, S. (2020). Exact neural mass model for synaptic-based working memory. PLoS Computational Biology 16, e1008533] and brain resting state activity in a large-scale brain network [Rabuffo, G., Fousek, J., Bernard, C., and Jirsa, V. (2020). Neuronal cascades shape whole-brain functional dynamics at rest. bioRxiv]. They have been also applied to describe abnormal Beta-Rebound in Schizophrenia [Byrne, Á., Coombes, S., & Liddle, P. F. (2019). A neural mass model for abnormal beta-rebound in schizophrenia. In Multiscale Models of Brain Disorders (pp. 21-27). Springer, Cham].

-Sec. 3.4 "Multiscale models"
There are specific models, here not mentioned, that have been constructed to model epileptic seizures. View their importance in the field, I think it is worth mentioning them in a review:

Jirsa, V. K., Stacey, W. C., Quilichini, P. P., Ivanov, A. I., and Bernard, C. (2014). On the nature of seizure dynamics. Brain 137, 2210-2230
Chizhov, A. V., Zefirov, A. V., Amakhin, D. V., Smirnova, E. Y., and Zaitsev, A. V. (2018). Minimal model of interictal and ictal discharges "epileptor-2". PLoS computational biology 14, e1006186

The former models have been used to construct in-silico approaches for the exploration of causal mechanisms of brain function and clinical hypothesis testing, see for example
Proix, T., Bartolomei, F., Guye, M., and Jirsa, V. K. (2017). Individual structural connectivity defines propagation networks in partial epilepsy. Brain 140, 641-654
Olmi, S., Petkoski, S., Guye, M., Bartolomei, F., and Jirsa, V. (2019). Controlling seizure propagation in large-scale brain networks. PLoS computational biology 15, e1006805

Minor issues
-Pag. 1, line 29  "to investigate one OF the most intriguing biological processes"

-Pag. 2, line 58 "profound ANALYSES of data" 

-Pag. 5, line 181 "neurotransmitter"

-Pag.7, lines 279-280. Please review the list "i) growth mechanisms, ii)
monomere addition of iii) both fibrils and filaments and iv) mass fraction and fibril length", because at the moment it is not clear which are the different cases.

Author Response

We thank the editor and reviewers for thorough comments and for raising critical issues to the original version of the manuscript.

Given the amount of changes that we have made in the text, we prefer not to reply point by point to reviewers’ concerns. Conversely, according to reviewers’ suggestion, the manuscript has been largely rewritten and restructured: sections are now organized and divided following a “pathologies” scheme. Concerns regarding the language have been considered, while new references and citations have been added taking into account reviewers’ suggestions.

Reviewer 2 Report

This paper provides a large catalog of computational models used in neuroscience and in studying neurological disorders. Unfortunately, it fails in some sense to really justify its final conclusions about the importance of computational models as tools to fight neurological disorders. I think the primary issue with this argument are that this paper is too broad and shallow. The main conclusions would be better argued for by presenting in greater depth a smaller number of examples where computational models played a pivotal role in discoveries. Or alternately, by performing some type of analysis of the growth of the work in the field. 

Furthermore, the use of english in this paper would benefit from editing by a native speaker. In the first paragraph, there are a number of items that strike this native speaker as odd. On line 29, saying "concurrently". line 32 "require to make assumptions". This pattern continues throughout. 

Based on these two major factors, this reviewer cannot recommend publication at this time. 

The following are some additional specific concerns the authors may wish to address. 

l 27. One might mention the span of these scales more specifically to make this point, like a certain number of spatial or temporal orders of magnitude. This general statement, while true, doesn't mean it is "necessary" to use multidiscriplinary approaches, whereas a larger number of orders of magnitude, would strengthen this argument. 

l. 37. Not clear what the difference is between structures and architectures. 

l. 45. "stochasticity". "randomness" would seem clearer and simpler. 

l. 48. Not clear why alterations due to pathologies cannot be explored experimentally. 

l. 50. Seems almost to be assuming the conclusion that the paper is designed to demonstrate. 

l. 58. 'Dynamic models' is a switch in terminology from 'pure simulations' on l. 56. 

l. 232. 'Alheimer's' is a possessive. 

l. 246. Are there examples of successful use of modeling in epilepsy? 

l. 255. Such as which investigation specifically -- citations needed here. 

l 272. spelling for dysfunction

Author Response

(The authors gave the same response as above.)

Round 2

Reviewer 1 Report

The authors have sufficiently improved the manuscript to make it publishable

Author Response

We thank the reviewer for the careful and thorough revision of the manuscript.

Reviewer 2 Report

The authors have made significant improvements to make this a better organized overview which is more informative. I would recommend publication.

One item which the authors may wish to consider changing is the use of "an exponential advancement". Normally exponential means that some dependent variable is growing as an exponential function of an independent variable. It is not clear what the 'exponential' independent variable would be here. I know this is a popular colloquial expression these days, but would suggest avoiding it in scientific writing. 

Author Response

We thank the reviewer for the thorough and careful revision of the manuscript.

According to reviewer suggestion, we have changed the last paragraph into the following sentences:

"The parallel progresses in the experimental and computational Neuroscience, together with the improvement of High-Performing Computing prelude to an exponential increase in the development of computational models aiming to investigate the mechanisms underlying neurological diseases. In the very next future, we can foresee that researches and scientists’ efforts will be increasingly devoted to the generation of computational models in order to provide effective therapeutic and pharmacological strategies"